# Human iPSC-Derived Cortical Neurons Display Homeostatic Plasticity

**DOI:** 10.3390/life12111884

**Published:** 2022-11-14

**Authors:** Federica Cordella, Laura Ferrucci, Chiara D’Antoni, Silvia Ghirga, Carlo Brighi, Alessandro Soloperto, Ylenia Gigante, Davide Ragozzino, Paola Bezzi, Silvia Di Angelantonio

**Affiliations:** 1Department of Physiology and Pharmacology, Sapienza University of Rome, 00185 Rome, Italy; 2Center for Life Nano- & Neuro-Science of Istituto Italiano di Tecnologia (IIT), 00161 Rome, Italy; 3CrestOptics S.p.A., Via di Torre Rossa 66, 00165 Rome, Italy; 4D-Tails s.r.l., Via di Torre Rossa 66, 00165 Rome, Italy; 5Santa Lucia Foundation, European Center for Brain Research, 00143 Rome, Italy; 6Department of Fundamental Neurosciences, University of Lausanne, 1015 Lausanne, Switzerland

**Keywords:** neurons, plasticity, iPSC, astrocytes, action potential, calcium imaging, synapse, neuronal networks, human, homeostatic plasticity, neuronal differentiation

## Abstract

**Simple Summary:**

In the brain, homeostatic plasticity is a form of synaptic plasticity in which neurons respond to chronically altered network activity with a negative feedback to maintain healthy brain function. In this way, when the activity of the network is chronically reduced, homeostatic plasticity normalizes all neural synaptic connections by increasing the strength of each synapse so that the relative synaptic weighting of each synapse is preserved. Here, we developed an in vitro human cortical model system, using human-induced pluripotent stem cells, and we demonstrated for the first time the presence of homeostatic plasticity in human neuronal networks. Since induced pluripotent stem cell-derived neurons can be obtained from patients with neurodevelopmental and neurodegenerative diseases, our platform offers a versatile model for assessing human neural plasticity under physiological and pathological conditions.

**Abstract:**

Maintaining the excitability of neurons and circuits is fundamental for healthy brain functions. The global compensatory increase in excitatory synaptic strength, in response to decreased activity, is one of the main homeostatic mechanisms responsible for such regulation. This type of plasticity has been extensively characterized in rodents in vivo and in vitro, but few data exist on human neurons maturation. We have generated an in vitro cortical model system, based on differentiated human-induced pluripotent stem cells, chronically treated with tetrodotoxin, to investigate homeostatic plasticity at different developmental stages. Our findings highlight the presence of homeostatic plasticity in human cortical networks and show that the changes in synaptic strength are due to both pre- and post-synaptic mechanisms. Pre-synaptic plasticity involves the potentiation of neurotransmitter release machinery, associated to an increase in synaptic vesicle proteins expression. At the post-synaptic level, we report an increase in the expression of post-synaptic density proteins, involved in glutamatergic receptor anchoring. These results extend our understanding of neuronal homeostasis and reveal the developmental regulation of its expression in human cortical networks. Since induced pluripotent stem cell-derived neurons can be obtained from patients with neurodevelopmental and neurodegenerative diseases, our platform offers a versatile model for assessing human neural plasticity under physiological and pathological conditions.

## 1. Introduction

During the course of a life, neural circuits can express different forms of plasticity by altering their synaptic structure and strength. These changes are critically dependent on experience-driven neuronal activity. Hebbian synaptic plasticity has long been considered the most important form of plasticity but, over the last 20 years, increasing interest has been aroused in homeostatic mechanisms. Indeed, they can explain how a neuronal network retains its baseline function despite having to face multiple challenges, such as developmental plasticity, learning, or injury. Hebbian plasticity can be described as a positive feedback mechanism that reinforces synaptic connections and includes long-term potentiation (LTP) and long-term depression (LTD). However, homeostatic plasticity behaves like a negative feedback mechanism, used by neurons to counteract excessive excitation or inhibition in response to persistent changes in network activity.

According to Turrigiano et al. [1], homeostatic synaptic scaling, observed in vitro and in vivo in rodent neocortical and hippocampal neurons, modifies the strength of the neuron’s excitatory synapses up or down in order to stabilize firing when the inputs are changed [1,2,3,4]. Despite the increasingly extensive literature concerning homeostatic synaptic plasticity at rodent central synapses, its mechanism(s) of expression (pre- or post-synaptic or both) and the role of glial cells are still unclear. Synaptic strength may be altered as a result of (possibly simultaneous) changes in the post-synaptic receptor accumulation, probability of pre-synaptic release, or the number of functional synaptic contacts. Some studies reported that the amplitude (but not the frequency) of glutamatergic currents can be modified by persistently altered neuronal activity [1,3,5], resulting in firing stabilization by means of a negative feedback. These post-synaptic changes are due to changed glutamate sensitivity [1,3] and membrane insertion of α-amino-3-hydroxy-5-methyl-4-isoxazolepropionic acid (AMPA) receptors [2,3]. Other reports describe little or no change in glutamatergic current amplitude but an increase in the event frequency [6,7], thus suggesting changes in the probability of release.

Synaptic homeostasis has been observed in different organisms, including Caenorhabditis elegans, Drosophila melanogaster, crustaceans, and rodents [8,9,10,11,12,13,14]. Given the physiological adaptive role of homeostatic plasticity, its disruption could give rise to neural excitability disorders. However, it is not clear whether there are direct associations with disease. Recent studies investigate the association between defects in homeostatic plasticity and the pathogenesis of various neurodevelopmental and neurodegenerative conditions, which may rely on mutations in disease susceptibility genes on both sides of synapses, likely orchestrating the homeostatic control of synaptic function [11,15,16,17].

A number of theories have been proposed to dissect the machinery of synaptic malfunction underlying neurodevelopmental disorders [16,18,19]. Among them, dysregulated homeostatic synaptic plasticity could cause network excitation imbalances, as described in mouse models of neurodevelopmental disorders [12,18,20,21,22,23]. However, animal models do not fully reflect development, genetics, and disease mechanisms that are unique to the human brain. Specifically, physiological and evolutionary species-specific differences so far hinder the translation of mechanistic findings concerning synaptic scaling from rodents to humans.

Over the last ten years, the discovery of adult cell reprogramming of induced pluripotent stem cell (iPSC) [24,25] has provided a previously unheard of opportunity for modeling neurodevelopment and its disorders, overcoming the difficulties of accessing the human brain [26,27,28,29,30].

The purpose of the present study was to investigate, on both sides of the synaptic cleft, the effects of blocking synaptic transmission, using a preparation of cultured human neocortical neurons differentiated from hiPSCs. We found significant synaptic functional alterations after two days of Tetrodotoxin (TTX) treatment, associated with changes in the expression of pre- and post-synaptic proteins. Our results suggest that homeostatic plasticity in the developing human excitatory neocortical synapses is a result of coordinated changes in the pre- and post-synaptic compartments that act as a gain control mechanism.

## 2. Materials and Methods

### 2.1. Human iPSC Maintenance and Differentiation into Cortical Neurons

hiPSC lines were cultured onto Matrigel functionalized plates (hESC-qualified, CORNING, Corning, NY, USA) in a medium containing Nutristem-XF (Biological Industries, Kibbutz Beit-Haemek, Israel) and 0.1% penicillin–streptomycin (Thermo Fisher Scientific, Waltham, MA, USA). The medium was refreshed daily, and the cells were passaged with Dispase II (1 mg/mL; Thermo Fisher Scientific) every 4–5 days.

As previously described, hiPSCs were differentiated into cortical neurons according to Shi et al, 2012 [31,32]. In brief, hiPSCs were treated with Accutase (Thermo Fisher Scientific), and the single-cell suspension was seeded into Matrigel-coated dishes (100,000 cells per cm^2^ seeding density) in Nutristem-XF plus ROCK inhibitor (10 μM). The medium was changed the day after seeding to N2B27 medium consisting of DMEM-F12, Dulbecco’s Modified Eagle’s Medium/Nutrient Mixture F-12 Ham (Merck Life Science, Milano; Italia), Neurobasal Medium (Thermo Fisher Scientific), 1X N2 supplement (Thermo Fisher Scientific), 1X GlutaMAX (Thermo Fisher Scientific), 1X MEM-NEAA (Thermo Fisher Scientific), 1X B27 (Thermo Fisher Scientific), and 1X penicillin–streptomycin (Thermo Fisher Scientific). We then considered this time point as day 0 (D0). At D0, we added SMAD inhibitors to induce neural fate (10 μM SB431542, 500 nM LDN-193189; Cayman Chemical, Ann Arbor, MI, USA). At D10, the neuroepithelial sheet was broken up into clumps of approximately 500 cells/clump and re-plated, in N2B27 medium, onto dishes coated with poly-L-ornithine/laminin (1X; Merck Life Science). At D20, cells were dissociated using Accutase and seeded onto poly-L-ornithine/laminin-coated dishes at 100,000 per cm^2^ in N2B27 medium supplemented with 10 μM ROCK inhibitor for 24 h. Culture medium was then changed every two days. At D27, Cyclopamine (2 μM; Merck Life Science) was added for 4 days to the N2B27 medium. From D30 to D70, immature neurons were re-plated into poly-L-ornithine/laminin-coated dishes at 100,000 per cm^2^ in N2B27 medium supplemented with BDNF (20 ng/mL; Sigma Aldrich), GDNF (20 ng/mL; Peprotech, Cranbury, NJ, USA), ascorbic acid (200 ng/mL; Sigma Aldrich), cyclic AMP (1 mM; Sigma Aldrich), and DAPT (5 μM; Adipogen Life Sciences; San Diego, CA, USA). Cells were routinely tested for mycoplasma contamination. To block network activity, TTX was added to the culture medium (1 µM, 48 h).

### 2.2. Immunostaining and Image Acquisition and Analysis of 2D Cultures

iPSC derived cortical cultures were fixed in paraformaldehyde (4% PFA solution; Merck Life Science) at room temperature for 15 min and then washed twice (1X PBS; Thermo Fisher Scientific). Fixed cells were then permeabilized using a PBS solution containing 0.2% Triton X-100 (Merck Life Science) for 15 min and blocked with the same solution supplemented with 5% goat serum (Merck Life Science) for 20 min at room temperature. Cells were then incubated with primary antibodies overnight at 4 °C using the following dilutions: mouse anti-PAX6 (sc81649 Santa Cruz Biotechnology, 1:50), chicken anti-MAP2 (ab5392 Abcam, 1:2000), rabbit anti-β-TUBULIN III (TUJ1) (T2200 Merck Life Science, 1:2000), mouse anti-GFAP (MAB360 Merck Life Science, 1:500), rabbit anti-PSD95 (3450 Cell Signaling, Danvers, MA; USA, 1:250), mouse anti-VGLUT1 (135303 Synaptic Systems, Göttingen, Germany, 1:250), rabbit anti-synapsin1 (D12G5 Cell Signaling, 1:200), and mouse anti-GAD67 (sc-28376 Santa Cruz, 1:250) [33]. The second day, cells were incubated with AlexaFluor secondary antibodies (1 h at room temperature; Thermo Fisher Scientific; 1:750); DAPI (Merck Life Science) was used for nuclei staining. Control experiments in the absence of primary antibody incubation were performed to check staining specificity.

Confocal fluorescence images were acquired with a 60×/NA 1.35 oil objective (Olympus, Tokyo, Japan) in stack with z-step of 0.5 μm using a microscope Olympus iX73 equipped with an LDI laser illuminator (89 North, Williston, VT, USA), an X-Light V3 spinning disc head (CrestOptics, Roma, Italy), a camera Evolve EMCCD (Photometrics, Tucson, AZ, USA), and the software MetaMorph (Molecular Devices, San Jose, CA, USA). Quantification was performed for each field of view (FOV; 400 µm^2^) with Fiji software using the “Puncta Analyzer” plugin. Stack images were merged in a maximum intensity Z projection and binarized to allow synaptic point counting and co-localized point quantification. The fused image with pre-synaptic staining of VGLUT1 (channel 1) and post-synaptic protein PSD95 (channel 2) was split, as required by the plugin, for threshold adjustment of each synaptic marker. The threshold was then set to eliminate the non-specific background signal for each channel. Finally, the quantification of synaptic dots for each channel and co-localized dots was performed by the “Puncta Analyzer” plugin.

### 2.3. PCR, RT-PCR and RT-qPCR

Genomic DNA was extracted from the cells using the PCRBIO Rapid Extract PCR Kit (PCR Biosystems, London, UK) according to the manufacturer instructions. After extraction, genomic DNA was amplified using oligos indicated in Table 1 for 40 cycles.

Total RNA was extracted from the cells with the EZNA Total RNA Kit I (Omega Bio-Tek, Norcross, GA, USA) and retrotranscribed using the Reverse Transcription Supermix for RT-qPCR iScript (Bio-Rad, Hercules, CA, USA).

The cDNA of the housekeeping gene ATP5O (ATP synthase, H+ transporting, mitochondrial F1 complex, O sub-unit) was used as internal control and amplified for 28 reaction cycles.

Real-time RT-PCR was performed with the iTaq Universal SYBR Green Supermix (Bio-Rad) using a ViiA 7 Real-Time PCR System (Applied Biosystems Waltham, MA, USA).

### 2.4. Calcium Imaging Recordings and Data Processing

Time lapse fluorescence images were recorded at room temperature using a custom digital imaging microscope. Excitation of Fluo4 calcium dye was obtained using the light source Lambda XL (Sutter Instrument, Novato, CA, USA) set at 488 nm wavelength with a Lambda 10-B optical filter changer (Sutter Instrument). A 525/50 nm filter was used to collect the emitted light. The emitted fluorescence was collected using a Zeiss Axio observer A1 inverted microscope (Zeiss, NY, USA) equipped with a Zeiss A-Plan 10×/NA 0.25 objective (Zeiss) and a CoolSNAPHQ2 camera (Photometrics). Image acquisition at sampling rate of 4 Hz was performed with Micromanager software. Changes in the intra-cellular calcium level were monitored with the high-affinity calcium-sensitive indicator Fluo4-AM (Thermo Fisher Scientific) used at a concentration of 5 μM by incubating neuronal cultures at 37 °C for 30 min in HEPES-buffered external solution containing 140 mM NaCl, 2.8 mM KCl, 2 mM CaCl^2^, 2 mM MgCl^2^, 10 mM HEPES, and 10 mM D-glucose (pH 7.38 with NaOH; 290 mOsm).

Calcium imaging data processing was performed using a custom MATLAB code [34]. This software automatically detected neuron positions analyzing the cumulative difference between frames in the time series, providing the ability to manually adjust any false or missing detections. Once the cell locations were established, their time-dependent fluorescence intensity traces were collected and examined to detect the occurrence of spike-associated calcium events.

Local peaks were identified by imposing thresholds for peak amplitude (1% of the baseline value), trace slope at onset and offset (10-3 and -10-4, respectively), and a minimum time interval within onset and offset (0.75 s) [35]. Each detected fluorescence transient was then fit in a two-step procedure with a model function consisting of a single-exponential rise and decay to obtain the shape of the signals (amplitude, rise time, and decay time) [36]. Analysis of calcium dynamics was critical to distinguish fast calcium transients, typical of neuronal activity, from calcium signals characterized by a slower onset (release from internal stores, calcium signals from other cell types). Based on our experimental conditions, we established a threshold of τ* = 2 s to recognize and discard non-neuronal signals: neurons whose average rise time was less than this threshold were selected, and all other signals were not considered. Inter-event interval, amplitude, and network synchrony (evaluated as the relative number of simultaneous events) were exported in Microsoft Excel or Prism9 for final statistics.

For the glutamate induced calcium activity, images were acquired with an exposure time of 200 ms at 1 Hz frequency for 15 min through a BX51WI microscope (objectives: LUMPlanF N 10×/0.10, air, and 40×/0.80, water immersion, Olympus). Fluo4-AM was excited with an Optoscan monochromator (Cairn Research, Facersham, UK) at 488 nm using a xenon lamp Optosource (Cairn Research). A borosilicated glass puffer pipette was filled with 2 mM Glutamate (Sigma Aldrich) in NES and moved via a MP-225 micro-manipulator (Sutter Instruments, Novato, CA, USA) to reach the core of the field of view, approximately 50 µm over the surface of the dish [37].

Basal fluorescence was assessed for 5 min; then a small volume of agonist-containing solution was puffed on the cells using a pneumatic pico-pump (PV820; World Precision Instruments, Inc., Sarasota, FL) with a short pressure (10 psi; 100 ms; [37,38,39]).

Images were acquired using a CCD CoolSnap MYO camera (Photometrics, Tucson, AZ, USA) and analyzed with the MetaFluor software as fluorescence variation into each region of interest (ROI) corresponding to single cells. To quantify the signal, the formula (F-F0)/F0 was used, where F0 is the mean fluorescence before agonist application, and F is the fluorescence intensity during the time-lapse acquisition.

### 2.5. Code Availability

For the analysis calcium imaging data, we used custom code based on MathWorks (Natick, MA, USA) Matlab (version 2016b) software, available upon request.

### 2.6. Statistical Data Analysis

All the statistical analysis, graphs, and plots were generated using MATLAB 2016b (MathWorks) and GraphPad Prism 9 (GraphPad Software, Insight Partners, NY, USA). To test whether our data sets reflected a normal distribution, the Shapiro–Wilk normality test was performed. In cases where the normality distribution was not satisfied, statistical significance analysis was performed using the non-parametric two-sided Mann–Whitney test (MW test, *p* < 0.05). In all other cases, unless otherwise specified, the *t*-Student test was used (*p* < 0.05). Data are shown as mean ± standard error of the mean (s.e.m.); the number of cells, replicates, fields of view, and differentiation batches are given for each experiments in the corresponding figure legend. Statistical significance of the cumulative distribution was calculated using the non-parametric Kolmogorov–Smirnov test.

## 3. Results

We used a hiPSC line derived from healthy subject fibroblasts [40], to obtain cortical cultures by means of a slightly modified conventional differentiation protocol under 2D culture conditions (Figure 1A). Acquisition of a cortical neural fate was induced by dual inhibition of SMAD, followed by blockade of Hedgehog signaling with cyclopamine [31]. Exit from pluripotency and acquisition of neuronal characteristics was assessed by the analyses of pluripotency (SOX2), neural progenitor cells (PAX6), and neuronal (TBR2) markers by qRT-PCR during the first 25 days of the differentiation protocol and then throughout the 70 days of culture maturation (Figure 1B, left).

The 70-day-long analysis showed the temporally regulated expression of neuronal (MAP2, TBR1) and astrocytic markers (GFAP), thus indicating the maturation of the cortical culture (Figure 1B, right). Immunofluorescence analysis at the neural rosette stage (day 20) confirmed the expression of PAX6. We then analyzed in at least three independent differentiation experiments a panel of early and late neural and glial markers. At day 25 (corresponding to the exit from the cell cycle and the acquisition of neuronal characteristics; [31]), we found a reduction in the PAX6 signal and the expression of the MAP2 neuronal cytoskeletal marker (see Figure 1C). The complexity of the network progressively increased up to day 50, with the presence of the TBR1 cortical neuronal marker and the initial expression of the GFAP glial marker. By day 70, cortical maturation was revealed by the high level of the GFAP astrocytic marker (Figure 1C). Taken together, these results validated the maturation of mixed neuronal and glial cortical cultures obtained from hiPSC line differentiation.

To evaluate the effects of neuronal inactivity on synaptic connectivity/maturation in the hiPSC-derived cortical cultures, we chronically blocked action potentials (APs) with TTX (1 µM, for 2 day) at two different developmental stages (day 50 and day 70). Forty-eight hours later, the effect of chronic TTX administration on the glutamatergic and GABAergic systems was evaluated in TTX-treated and control sister cultures (*n* = 3 batches).

We first analyzed the role of AP-mediated network activity on day 50, a differentiation stage characterized, in our culture system, by the presence of both glutamatergic and GABAergic neurons (see below) and by the early presence of astrocytes (see Figure 1B). Confocal immunofluorescence analysis showed that TTX-mediated network inactivation significantly increased post-synaptic density (PSD95, Figure 2A), in line with published findings in rodent cortical primary cultures [41]. Analysis of the expression of the broad pre-synaptic marker SYN1 (Figure 2B) revealed a significant TTX-mediated increase limited to glutamatergic synapses. Indeed, AP blockade increased VGLUT1 levels (Figure 2C), leaving unaltered the expression of the GABAergic marker GAD67 (Appendix A). Strikingly, AP blockade increased the co-localization of pre- (VGLUT1) and post-synaptic (PSD95) glutamatergic markers on day 50 (Figure 2D), indicating that TTX treatment promotes positive homeostatic plasticity and faster glutamatergic synaptic maturation.

We then analyzed the impact of chronic TTX treatment at a more mature stage of cortical network development (DAY 70), when the substantial presence of astrocytic cells was revealed by high level of GFAP transcript (Figure 1B) and confirmed by strong GFAP staining (Figure 1C) [31]. Comparison of control and TTX-treated cultures showed that the AP blockade positively modulated the expression of pre- (VGLUT1) and post-synaptic (PSD95) glutamatergic proteins (Figure 3A,C), without having a broad effect on synaptic vesicle protein SYN1 (Figure 3B). The positive homeostatic plasticity of glutamatergic synaptic contacts was confirmed by the enhanced co-localization of VGLUT1 and PSD95 (Figure 3D), thus suggesting that the AP blockade increased the connectivity of the glutamatergic network even at day 70. Conversely, TTX treatment did not change the expression of pre-synaptic GABAergic specialization (GAD67) (Appendix A), suggesting the lack of homeostatic plasticity of inhibitory transmission also at this stage of network maturation.

It is particularly noteworthy that the homeostatic plasticity observed at day 70 was also supported by the measurements of spontaneous intra-cellular calcium transients and glutamate-induced calcium responses at the single neuron level. Spontaneous calcium dynamics in hiPSC-derived cortical neuronal networks loaded with Fluo-4M were recorded for 5–10 min and analyzed using a custom code [31] that allowed us to (i) recognize cells; (ii) select active cells; (iii) sort calcium events; (iv) divide cells in two groups, based on the rise time of calcium events; and (v) extract functional properties, such as amplitude, frequency, kinetic parameters, and network synchronicity in control a TTX-treated cultures.

The analysis indicated that TTX treatment significantly increased spontaneous activity in hiPSC-derived cortical neurons (characterized by fast events), inducing in treated cultures higher amplitude of calcium transients (Figure 4A,C) and shorter inter-event intervals (Figure 4A,C), compared to controls. Conversely, we did not observe differences in synchronized firing index or in the number of active neurons (Figure 4B).

Because the activity and excitability of the cortical network are closely dependent on the balance of excitatory and inhibitory transmission, we also evaluated the calcium transients evoked by the activation of glutamatergic transmission, by local application of glutamate, via a puffer pipette (2 mM, 100 ms, 10 p.s.i.). The analysis confirmed the functional expression of glutamate receptors in cultured neurons as a localized puff of glutamate evoked rapid intra-cellular calcium responses (Figure 4D). Interestingly, the TTX-treated neurons showed significantly greater glutamatergic responses (Figure 4D) than the controls, thus paralleling the observed increase in post-synaptic markers.

Taken together, the above data suggest that homeostatic plasticity induced by network silencing can be observed at developing synapses in human cortical networks, affecting both the pre- and the post-synaptic compartments.

## 4. Discussion

In this study, we created a platform enabling the analysis of homeostatic neural plasticity in cultured human neurons during maturation by differentiating hiPSCs into cortical cells and following their network properties for up to 70 days and found that homeostatic plasticity can be efficiently elicited by blocking the APs of human neuronal networks. We also found that homeostatic plasticity occurs at both pre- and post-synaptic levels in hiPSC-derived cortical neuronal networks, which underscores the utility of our in vitro platform to examine the plasticity of human neural networks under physiological and pathological conditions.

Homeostatic plasticity has been reported to be involved in the pathophysiology of several neurodevelopmental and neurodegenerative disorders, including intellectual disability [42], Rett syndrome [17], schizophrenia [43], and Alzheimer’s disease [44]. Synaptic scaling is a form of homeostatic plasticity described in cultured rodent brain slices, likely underlying brain network function, development, and plasticity. However, to the best of our knowledge, it has not yet been demonstrated in human cortical cultures.

This makes it even more challenging to develop effective drugs for neurodevelopmental disorders, which is already hampered by our limited understanding of their pathophysiology, differences between patients and animal models, and difficulties in modeling human brain development in vitro and in vivo. However, advances in the generation of disease-relevant hiPSCs and their modification through genome editing have provided new opportunities for disease modeling and drug screening for rare diseases.

In this framework, the lack of homeostatic plasticity has been reported in an in vitro model of fragile X syndrome based on patient-derived iPSCs [45]. However, it should be considered that this model is based on a human–mouse co-culture system that does not reflect the processes of cell differentiation and maturation that occur in human cortical tissue.

Indeed, the authors obtained a sub-population of pure excitatory cortical neurons from human iPSCs and embryonic stem cells through overexpression of the human NGN2 gene [45,46]. Then, to force synaptic maturation, mixed cortical and hippocampal primary cultures obtained from post-natal mice were added to the NGN2 neurons during the differentiation process. In this case, we decided to use a differentiation protocol based on dual inhibition of SMAD, followed by blockade of the Hedgehog signal, to better recapitulate cortical development [47] for both neuronal and glial components.

The locus of expression of homeostatic plasticity has significant functional implications. Pre-synaptic changes in the probability of neurotransmitter release profoundly alter short-term synaptic plasticity [48], and it is predicted that such changes in the dynamics of synaptic transmission alter information flows in central circuits [49,50], whereas post-synaptic changes in receptor accumulation can reset post-synaptic activity without influencing the short-term dynamics of synaptic transmission.

A major strength of our system is that human cortical cultures are characterized by the presence of homeostatic plasticity of glutamatergic synaptic events during early and late maturation (day 50 and 70, respectively) and this occurs both pre- and post-synaptically. In addition to observing enhanced network activity after TTX treatment, we found an increased calcium response to the focal application of glutamate, possibly associated to up-regulation of glutamatergic receptors. However, we cannot exclude the involvement of other downstream mechanisms, such as voltage-activated calcium channels [51].

It should be noted, however, that unlike what was observed at day 50, TTX treatment increased VGLUT1 staining at day 70 without affecting the expression of SYN1 or GAD67 markers. We can hypothesize that the absence of TTX effect on SYN1 at day 50 is due to the presence of other neurons in the culture, not undergoing homeostatic plasticity, expressing neither VGLU1 nor GAD67. This may be due to maturation in culture of iPSCs, resulting in developmentally regulated switching of VGLUTs isoforms [52], or differentiation into neurons not strongly associated with the glutamatergic or GABAergic phenotype [53].

For a complete understanding of the mechanism of homeostatic plasticity, it should be also discussed the possible role of astrocytes, significantly present in mature hiPSC-derived cortical cultures [31]. It is becoming increasingly clear that glial cells, and in particular astrocytes, play a key role in various functions within the central nervous system (CNS) in the physiological processes of the developing and mature brain [54,55,56,57,58,59,60,61,62,63,64] and take part in many key pathophysiological functions, such as neurotransmitter release, cell survival, mitochondria biogenesis, vascular tone, apoptosis, and gene expression [65,66,67,68,69,70,71,72]. Astrocytes can modulate synaptic transmission [61,62,64,73] and play a role in homeostatic plasticity since the coordination of the cellular events involved in different types of synaptic plasticity requires not only the bi-directional communication between pre- and post-synaptic neurons but also the interactions between neurons and astrocytes in tripartite structures [41,54,55,56,66,67,68,69,70]. Indeed, the TTX-induced blockade of neuronal activity in cultured hippocampal cells increases the expression and secretion of TNFα, which in turn modulates the homeostatic plasticity of both inhibitory and excitatory neurons [68,73,74,75,76,77,78,79,80]. Moreover, the astrocyte secretion of interleukin 33 is selectively regulated in order to maintain network homeostasis during hippocampal homeostatic synaptic plasticity [81]. In our experimental settings, it is easy to assume that astrocytes contribute to the development of homeostatic plasticity at day 70, when the percentage of astrocytes in our culture is consistent with mature cortical cultures. Regarding the nature of this contribution at day 50, we have to consider that GFAP expression is significantly lower (about 20% compared with day 70), as revealed by real-time PCR analysis, consistent with an early maturation phase. However, we cannot exclude that these astrocytes also contribute to homeostatic plasticity at day 50.

In brief, we used our hiPSC-based cortical differentiation platform to recapitulate the development of human cortical networks and demonstrated the presence of homeostatic plasticity during different phases of network development. Our long-term goals are to overcome the problems due to differences between humans and animal models, acquire insights into the functional properties of human neuronal development, and assess human neural plasticity under physiological and pathological conditions.

## Figures and Tables

**Figure 1 life-12-01884-f001:**
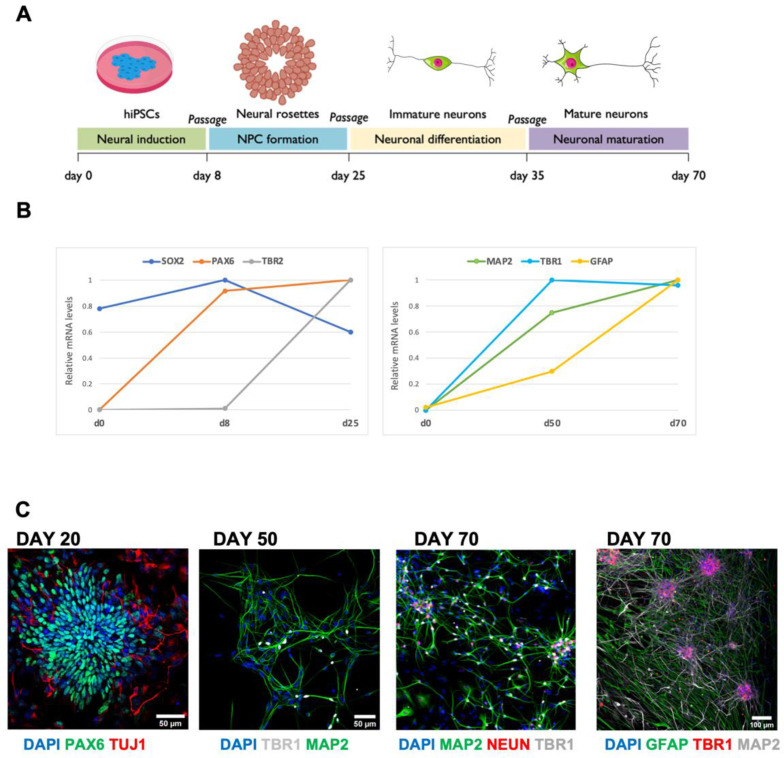
(**A**) Schematic representation of cortical differentiation timeline. (**B**) Time courses of the expression of neuronal and glial markers analyzed by real-time qRT-PCR in differentiating hiPSC cells. (**C**) Representative images of immunostaining for neural progenitors, cortical neurons, and astrocytes during the differentiation process: PAX6 (green) and TuJ1 (red) at day 20; TBR1 (gray) and MAP2 (green) at day 50; MAP2 (green), NEUN (red), and TBR1 (gray) at day 70; TBR1 (red), Scale bar: 50 µm; MAP2 (gray) and GFAP (green) at day 70, Scale bar: 100 µm. DAPI (blue) for nuclei visualization. (*n* = 2 differentiation batches). Note the increased network complexity and glial maturation at day 70. The scheme in A was generated using the Servier Medical Art, provided by Servier, under a Creative Commons Attribution 3.0 unported license.

**Figure 2 life-12-01884-f002:**
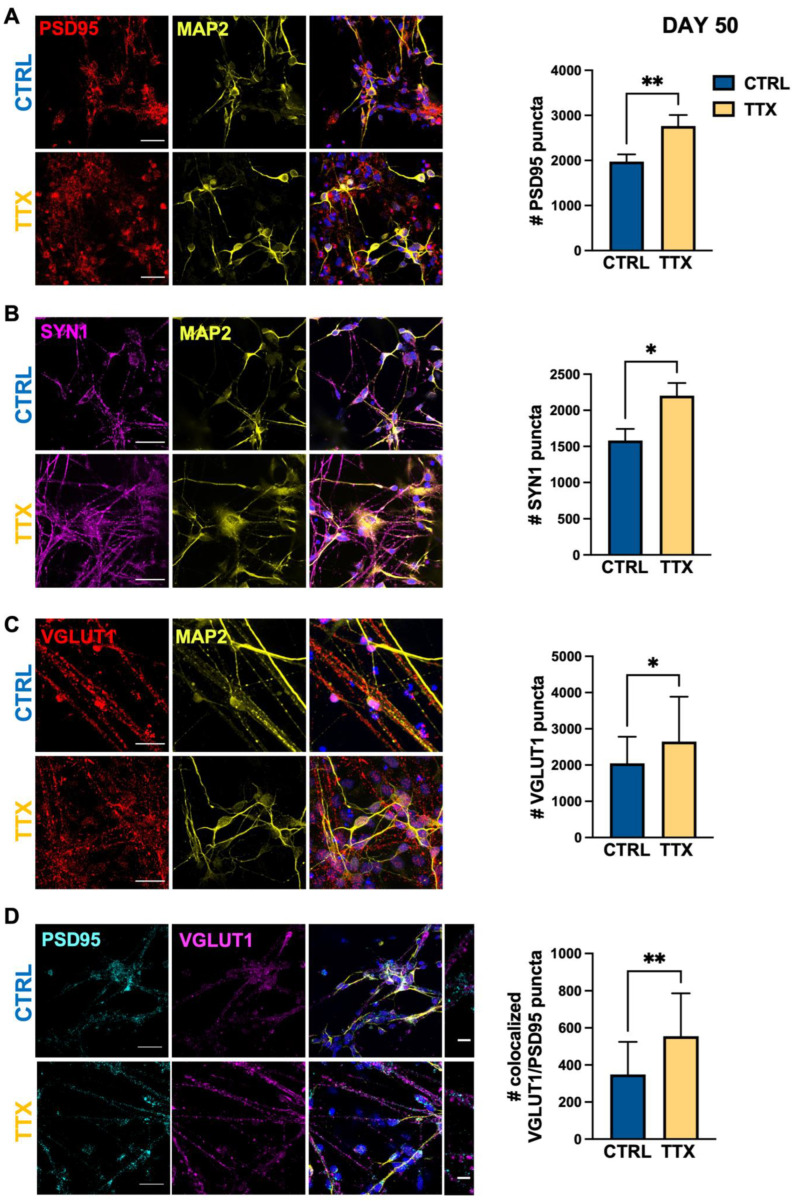
TTX induced homeostatic plasticity glutamatergic synapses at day 50. Representative immunofluorescence images (left panels) of synaptic proteins at day 50 in cortical cultures differentiated from hiPSCs in control condition (CTRL; blue) and after 48 h of TTX treatment (TTX; yellow) and relative bar chart for staining quantification (right panels). (**A**) Post-synaptic glutamatergic synapses are identified as positive for PSD95 (post-synaptic, red) on neurons stained for the dendritic cytoskeletal protein MAP2 (yellow) (PSD95 CTRL 1980 ± 150, TTX 2770 ± 240, ** *p* < 0.01, MW test; *n* = 26/2 FOVs; differentiation batches). (**B**) Pre-synaptic vesicles are identified as positively stained for SYN1 (magenta) and neurons as MAP2 positive cells (yellow) per FOV (400 μm^2^) (SYN1 CTRL 1580 ± 160, TTX 2200 ± 170, * *p* < 0.05, MW test; *n* = 25/2 FOVs; differentiation batches). (**C**) Glutamatergic post-synaptic puncta are identified as positive for VGLUT1 (pre-synaptic, red); MAP2 (yellow) as neuronal marker. (VGLUT1 CTRL 2050 ± 130, TTX 2650 ± 280, * *p* < 0.05, t-Student test; *n* = 26/2 FOVs; differentiation batches; FOV 400 μm^2^). (**D**) Glutamatergic synapses are identified as co-localized positive puncta per FOV (400 µm^2^) for both pre- (VGLUT1 magenta) and post-synaptic (PSD95 cyan) markers; MAP2 (yellow) for dendritic staining (VGLUT1/PSD95 co-localized puncta CTRL 350 ± 30, TTX 560 ± 50, ** *p* < 0.01 MW test; *n* = 30/2 FOVs; differentiation batches for each genotype). Data are expressed as mean ± s.e.m. Nuclei were stained with DAPI (blue); scale bars: 25 µm and 5 µm for zoomed images.

**Figure 3 life-12-01884-f003:**
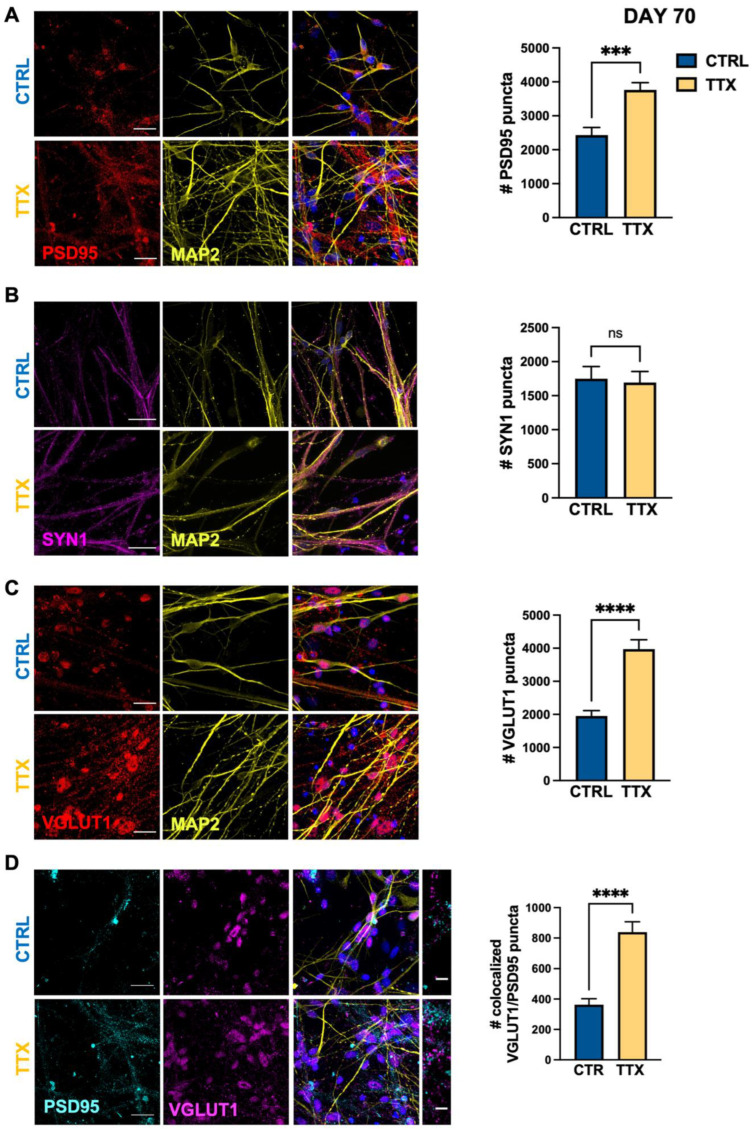
TTX induced homeostatic plasticity of glutamatergic synapses at day 70. Representative immunofluorescence images (left panels) of synaptic proteins at day 70 in cortical cultures differentiated from hiPSCs in control condition (CTRL; blue) and after 48 h of TTX treatment (TTX; yellow) and relative bar chart for staining quantification (right panels). (**A**) Post-synaptic glutamatergic synapses are identified as positive for PSD95 (post-synaptic, red) on neurons stained for the dendritic cytoskeletal protein MAP2 (yellow) (PSD95 CTRL 2430 ± 220, TTX 3760 ± 210, *** *p* < 0.001, MW test; *n* = 24/2 FOVs; differentiation batches; FOV 400 μm^2^). (**B**) Pre-synaptic vesicles are identified as positively stained for SYN1 (magenta) and neurons as MAP2 positive cells (yellow) (SYN1 CTRL 1750 ± 180, TTX 1690 ± 160, *p* = 0.82, t-Student test; *n* = 20/2 FOVs; differentiation batches FOV 400 μm^2^). (**C**) Glutamatergic post-synaptic puncta in each FOV (400 μm^2^) are identified as positive for VGLUT1 (pre-synaptic, red); MAP2 (yellow) as neuronal marker. (VGLUT1 CTRL 1950 ± 160, TTX 3970 ± 290, **** *p* < 0.0001, MW test; *n* = 25/2 FOVs; differentiation batches; FOV 400 µm^2^). (**D**) Glutamatergic synapses are identified as co-localized positive puncta for both pre- (VGLUT1, magenta) and post-synaptic (PSD95, cyan) markers; MAP2 (yellow) for dendritic staining (VGLUT1/PSD95 co-localized puncta CTRL 360 ± 40, TTX 840 ± 70, **** *p* < 0.0001 MW test; *n* = 25/2 FOVs; differentiation batches for each genotype; FOV 400 μm^2^). Data are expressed as mean ± s.e.m. Nuclei were stained with DAPI (blue); scale bars: 25 µm and 5 µm for zoomed images.

**Figure 4 life-12-01884-f004:**
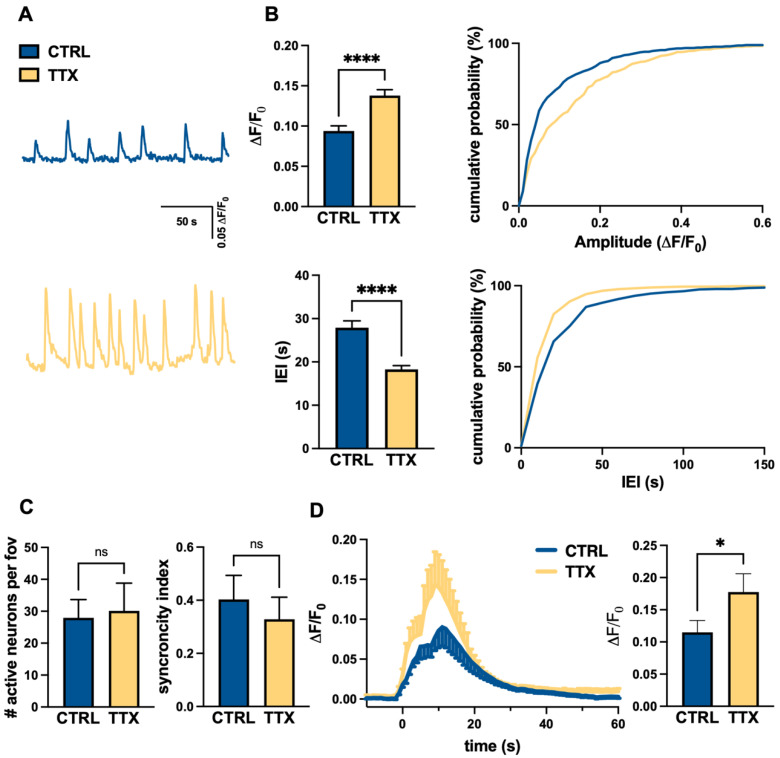
TTX induced homeostatic plasticity of glutamatergic spontaneous and evoked calcium activity in human iPSC-derived cortical cultures at day 70. (**A**) Representative calcium traces (ΔF/F0) of neurons (fast active cells) in control culture conditions (CTRL; blue, top trace) and after 48 h of APs blockages (TTX; yellow, bottom trace). (**B**) Analysis of the functional properties of active neurons in control (CTRL; blue) and TTX-treated (TTX; yellow) cultures. Top: bar charts representing amplitude and relative cumulative distribution of spontaneous calcium events (*n* = 14/2 FOVs/batches, **** *p* < 0.0001 CTRL vs. TTX, *t*-Student test); bottom: bar charts representing inter-event interval and relative cumulative distribution of spontaneous calcium events (*n* = 14/2 FOVs/batches, *p* < 0.0001 CTRL vs. TTX, *t*-Student test). (**C**) Bar charts representing the number active neurons for each FOV (left) and the synchronicity index (right) in both CTRL and TTX-treated cultures (*n* = 14/2 FOVs/batches for condition; CTRL 27 ± 6, TTX 30 ± 8 active cells *p* = 0.21, *t*-test; synchronicity index CTRL 0.40 ± 0.09, TTX 0.33 ± 0.08; *p* = 0.54, *t*-Student test) (**D**) Time course of the mean calcium transient response of control (blue) and TTX-treated (yellow) cortical neurons in response to puff application of 2 mM glutamate (CTRL *n* = 82/8/2 cells/FOVs/batches, TTX *n* = 77/8/2 cells/FOVs/batches); bar chart representing the mean peak amplitude value (ΔF/F0) for CTRL and TTX (* *p* < 0.05; MW test)-treated cells.

**Table 1 life-12-01884-t001:** List of primers used.

*Primer Name*	*Primer Sequence 5′-3′*
ATP5O FW	ACTCGGGTTTGACCTACAGC
ATP5O RV	GGTACTGAAGCATCGCACCT
GFAP FW	GATCAACTCACCGCCAACAG
GFAP RV	ATAGGCAGCCAGGTTGTTCT
MAP2 FW	TTCCTCCATTCTCCCTCCTCGG
MAP2 RV	TCTTCCCTGCTCTGCGAATTGG
PAX6 FW	ATGTGTGAGTAAAATTCTGGGCA
PAX6 RV	GCTTACAACTTCTGGAGTCGCTA
SOX2 FW	TCAGGAGTTGTCAAGGCAGAGAA
SOX2 RV	GCCGCCGCCGATGATTGTTATTA
TBR1 FW	GGAGCTTCAAATAACAATGGGC
TBR1 RV	GAGTCTCAGGGAAAGTGAACG
TBR2 FW	CTTCTTCCCGGAGCCCTTTGTC
TBR2 RV	TTCGCTCTGTTGGGGTGAAAGG

## Data Availability

The data that support the findings of this study are available from the corresponding author upon reasonable request.

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
