# Peer review of "Human iPSC-Derived Cortical Neurons Display Homeostatic Plasticity"

_life, 2022, doi:10.3390/life12111884_

Round 1

Reviewer 1 Report

In this study, Cordella et al. demonstrate the ability of human cortical neurons derived from iPSC ot display homeostatic synaptic plasticity in response to prolonged pharamcological blockade of activity with TTX. The study is logically conducted and clearly presented. However, I have important concerns that should be adressed:

1. In contrast to what is announced by the authors, the ability of human neurons to undergo homeostatic synaptic plasticity has been described previously, at least by one group, in particular in the context of fragile X syndrome (Zhang et al., Science Translational Medicine, 2018). Therefore, the authors should discuss their results in light of this previous study and cite it.

2. In Figure 2 and 3, the authors should provide high magnification images with good resolution to illustrate better the immunostainings so that the reader can appreciate neuronal morphology. In particular they should highlight synaptic puncta stained with PSD-95 and VGLUT1 antibodies to illustrate the changes induced by the TTX treatment.

3. In Figure 2 and 3, it is not clear what were the quantifications: density of puncta? average fluorescence intensity at each puncta? Size of the puncta? The authors should detail more precisely how they analysed the ICCs using the imageJ plugin.

4. In their calcium imaging experiments, it seems that the authors analyzed 'spike-associated calcium events'. Because spikes can results from non-linear integration at dendritic and somatic levels, it would be interesting to analyse spontaneous synaptic (local) events resulting from synapse activation (at dendritic level).

5. The authors do not provide evidence that the homeostatic synaptic plasticity they observe is uniform and affects all synapses equally, i.e., with a same gain. Yet, recent studies suggest that uniform homeostatic plasticity (aka 'synaptic scaling') may not be the norm (see Hanes et al., J Neurosci 2021; Dubes et al., EMBO J 2022; Wang et al., Neuropharmacology 2019). Therefore, the terms 'synaptic scaling' should be changed for 'homeostatic synaptic plasticity' throughout the manuscript.

6. The authors should propose an explanation to the fact that neurons display homeostatic plasticity (increased PSD-95, VGLUT1 and SYN1) even when astrocytes are virtually absent (i.e., at DIV 50). This is at odds with previous studies revealing an important role of astrocytes in homeostatic plasticity (see for instance Stellwagen and Malenka, Nature 2006), as mentioned in the discussion.

7. The authors should provide some explanation for the absence of effect on SYN1 at DIV70, while VGLUT1 is increased.

Reviewer 2 Report

This manuscript investigated the homeostatic plasticity in human iPSC derived cortical neurons. They found glutamate synaptic functional alterations after TTX treatment. Overall, this study provided some original evidence. The presentation is good. I just wondered if there are any new evidence on pathological disease models.
